

# Morphology and development of a novel murine skeletal dysplasia

Marta Marchini[1,2], Elizabeth Silva Hernandez[3] and Campbell Rolian[2,3]

[1] Department of Cell Biology and Anatomy, Cumming School of Medicine, University of Calgary, Calgary, Canada
[2] McCaig Institute for Bone and Joint Health, University of Calgary, Calgary, Canada
[3] Department of Comparative Biology and Experimental Medicine, Faculty of Veterinary Medicine, University of Calgary, Calgary, Canada

## ABSTRACT

**Background**. Limb bones develop and grow by endochondral ossification, which is regulated by specific cell and molecular pathways. Changes in one or more of these pathways can have severe effects on normal skeletal development, leading to skeletal dysplasias. Many skeletal dysplasias are known to result from mis-expression of major genes involved in skeletal development, but the etiology of many skeletal dysplasias remains unknown. We investigated the morphology and development of a mouse line with an uncharacterized mutation exhibiting a skeletal dysplasia-like phenotype (*Nabo*).
**Methods**. We used µCT scanning and histology to comprehensively characterize the phenotype and its development, and to determine the developmental stage when this phenotype first appears.
**Results**. *Nabo* mice have shorter limb elements compared to wildtype mice, while clavicles and dermal bones of the skull are not affected. *Nabo* embryos at embryonic stage E14 show shorter limb cartilage condensations. The tibial growth plate in *Nabo* mice is wider than in wildtype, particularly in the proliferative zone, however proliferative chondrocytes show less activity than wildtype mice. Cell proliferation assays and immunohistochemistry against the chondrogenic marker Sox9 suggest relatively lower, spatially-restricted, chondrocyte proliferation activity in *Nabo*. Bone volume and trabecular thickness in *Nabo* tibiae are also decreased compared to wildtype.
**Discussion**. Our data suggest that the *Nabo* mutation affects endochondral ossification only, with the strongest effects manifesting in more proximal limb structures. The phenotype appears before embryonic stage E14, suggesting that outgrowth and patterning processes may be affected. *Nabo* mice present a combination of skeletal dysplasia-like characteristics not present in any known skeletal dysplasia. Further genomic and molecular analysis will help to identify the genetic basis and precise developmental pathways involved in this unique skeletal dysplasia.

## INTRODUCTION

The mammalian skeleton develops primarily through two mechanisms: endochondral ossification and intramembranous ossification. In endochondral ossification, a developing bone is preceded by formation of a cartilaginous model, or anlage, within which

Corresponding authors
Marta Marchini,
mmarchin@ucalgary.ca
Campbell Rolian,
cprolian@ucalgary.ca

mesenchymal cells differentiate into chondrocytes (*Eames, De la Fuente & Helms, 2003*). This cartilage anlage is then replaced with bone, which grows longitudinally under the control of the growth plate (*Kronenberg, 2003*). In intramembranous ossification, direct differentiation of osteoblasts from mesenchymal cells results in direct apposition of bone without a cartilaginous model (*Hall, 2005*). Disruption of these mechanisms can produce severe phenotypes with defects in both endochondral and intramembranous ossification (*Eames, De la Fuente & Helms, 2003*; *Krakow & Rimoin, 2010*). Clinically, these phenotypes are collectively known as skeletal dysplasias (*Rimoin et al., 2007*).

Skeletal dysplasias (SDs) disrupt normal formation and development of bone (osteodysplasia), cartilage (chondrodysplasia), or both (*Beals & Horton, 1995*). As SDs are often the result of disruptions in major signaling pathways (*Rimoin et al., 2007*), SDs can show abnormal phenotypes in multiple tissues, alone or in combination, including limb and skull defects (*Ballock & O'Keefe, 2003*; *Shah, Varghese & Fernandes, 2017*), kyphosis, scoliosis and other vertebral anomalies (*Shirley & Ain, 2012*), as well as defects in other organ systems (*Krakow & Rimoin, 2010*). Together, these disorders affect between 2 and 4 live births per 10,000 (*Orioli, Castilla & Barbosa-Neto, 1986*; *Stoll et al., 1989*) with high perinatal mortality rates (*Barbosa-Buck et al., 2012*), short life expectancy, and lifelong disability (*Baker et al., 1997*), although long term prognosis depends on the type of dysplasia (*Krakow & Rimoin, 2010*). The genetic cause of many dysplasias remains unknown (*Srinivas & Shapiro, 2012*).

During an experiment selecting for increased tibia length in mice (*Marchini et al., 2014*), we discovered mice with a previously unreported short-limbed phenotype. Strong selection in relatively small populations can increase the frequency of rare genetic variants, leading to higher proportions of homozygotes (*Marsden et al., 2016*). We believe that this may have facilitated the appearance of this short limb phenotype. In this study, we investigate the morphology, bone and cartilage histology, and growth of this short-limbed mouse line, and discuss the potential cell mechanisms of bone and cartilage growth that lead to this peculiar phenotype.

# MATERIAL AND METHODS

## Sample collection

All experimental procedures were approved by the Health Sciences Animal Care Committee at the University of Calgary (protocol AC13-0077), and conducted in accordance with best practices outlined by the Canadian Council on Animal Care. The *Nabo* mouse, and the Longshanks mouse it originated from, are derived from Hsd-ICR (CD-1®) outbred stock (Harlan Biosciences, Indianapolis, IN). Mice are housed in individually ventilated cages (Greenline Sealsafe PLUS, Buguggiate, Italy), kept on a 12-hour light/dark cycle at a constant room temperature (20 °C), with food (Pico-Vac Mouse Diet 5061, LabDiet, St. Louis, MO) and water provided ad libitum.

*Nabo* first appeared in generation F14 of the Longshanks selective breeding experiment. This selection experiment consists of three independent lines of CD-1 mice bred under similar conditions for the same number of generations: two lines selectively bred for

long tibiae relative to body mass (Longshanks 1 and Longshanks 2), and a random-bred control line (wildtype) (*Marchini et al., 2014*). Longshanks mice at generation F14 showed an increase of 9–13% in tibia length relative to body mass over wildtype. For additional information see *Marchini et al. (2014)*. We discovered four siblings (two females and two males) from a litter of eight (line Longshanks 2, family O2) with short limbs. After a generation (F15), we identified two more females with a similar phenotype in a related family (line Longshanks 2, family H2). We named this phenotype and these mice *Nabo*. By mating a *Nabo* O2 male and *Nabo* H2 female, we were able to generate a litter (family Nabo01) of three mice (see pedigree, Fig. S1). We then paired a *Nabo* H2 female with a wildtype male and two Nabo01 males with wildtype females. This produced three litters of mice without the *Nabo* phenotype (called Nabo1A, Nabo3A and Nabo4A). We crossbred these mice to produce litters of mice with approximately 10% *Nabo* mouse phenotype, while the remaining mice were without phenotype (Fig. S1). *Nabo* mice were then inbred to obtain litters with exclusively *Nabo* phenotype. For all analyses, we used wildtype from the Longshanks experiment, and *Nabo* mice derived from Longshanks 2 (*Marchini et al., 2014*). Mice at selected ages were euthanized and stored at −20 °C or immediately processed as needed (see below).

## Postnatal growth curve and μCT scanning

To determine the extent, timing and pattern of skeletal growth retardation in *Nabo* mice, we used μCT scanning to measure the length of skull, clavicle, scapula, humerus, ulna, 5th metacarpal, femur, tibia and 5th metatarsal of five *Nabo* and five wildtype mice for each postnatal (P) age P0, P3, P7, P9/P10, P14, P21, P28, P42, P75, P100, and P125. Whole carcasses were thawed at room temperature and scanned using a Skyscan 1173 μCT scanner (Bruker, Kontich, Belgium), using appropriate scan parameters for the size of each mouse (voxel size range 30 μm–71 μm, X-ray 70–80 kV, 80–100 μA, no filter). Scans were reconstructed as tomographic stacks using NRecon v1.6.9 (Bruker, Kontich, Belgium) and linear measurements were taken via 3D landmarking in Amira v.5.4.2 (Visage Imaging, Berlin, Germany) as described (*Cosman, Sparrow & Rolian, 2016*; *Farooq et al., 2017*). Growth curves for each element were obtained by fitting the cross-sectional data with Gompertz logistic growth functions (*German et al., 1994*). Growth rates were obtained by taking the first derivative of the fitted growth curves (*Farooq et al., 2017*; *Marchini & Rolian, 2018*).

To assess whether endochondral and/or intramembranous elements in the skull were affected, we μCT scanned five P75 *Nabo* and five age-matched wildtypes (three males and two females in each group) (voxel size 45 μm, X-ray 70–80 kV, 80–100 μA) and took the following linear measurements of individual skull elements: length and width of the occipital and palatine bones, length of parietal, frontal, nasal bones, sphenoid and total length of the skull from the foramen magnum to the rostral tip of the nasal bone (Fig. S3). Linear measurements were made via 3D landmarking in Amira v.5.4.2.

## Embryonic growth

To investigate when in development the *Nabo* phenotype is first apparent, we measured the humerus, ulna, femur and tibia of *Nabo* and wildtype embryos ($n = 5 - 6$ each) at

developmental stages E14 and E18. For E14, we used iodine contrast-enhanced μCT scanning (*Gignac et al., 2016*), for E18 we used whole-mount Alcian blue/Alizarin red staining.

E14 embryos were harvested, placed in 1x PBS on ice and fixed in 10% neutral-buffered formalin. After fixation, embryos were placed in Lugol's iodine stain (0.375% $I_2$ and 0.75% KI in $dH_2O$). Lugol's iodine stains soft tissues for visualization with μCT scanning in a time and concentration dependent manner (*Gignac et al., 2016*), but does not stain cartilage, leaving a negative space on 3D renderings (Fig. S3), which can be digitally segmented. After several days, embryos were embedded in 1% agarose and scanned using the Skyscan 1173 μCt scanner (voxel size 7 μm, X-ray 65–70 KV; 85–96 μA, no filter). Scans were reconstructed as above. We then volumized the humerus, ulna, femur and tibia (Fig. S3) and calculated linear measurements from the resulting surface models by placing landmarks on their proximal and distal ends via 3D landmarking as above. To standardize cartilage measurements, we took linear measurements of the mediolateral width of the nose (nostril to nostril) and the mean interorbital distance and used them as covariates in our statistical analyses. To assess qualitatively any cellular disruption of the cartilage anlagen in the hind limb, we collected hind limbs of *Nabo* and an age-matched wildtype at embryonic stage E14.5 (Fig. S4). Hind limbs were fixed in 4% paraformaldehyde (Sigma) overnight at 4 °C. Samples were then dehydrated, embedded in paraffin, and sectioned in the coronal plane at 4 μm. Sections were deparaffinized in xylene, stained with Mayer's hematoxylin (Sigma), 1% Alcian blue in 1% acetic acid (Sigma), 1% Phosphomolybdic acid (Sigma), 0.5% Sirius red (Direct Red 80, Sigma) in 0.5% acetic acid and then cover slipped. The stained sections were imaged using a digital microscope (Axioplan 2, Zeiss) with attached camera (Optronics), using StereoInvestigator v7.

E18 fetal mice were harvested and placed in PBS on ice. The embryos were then skinned and internal organs were removed. The mice were then placed in ice-cold 95% ethanol for one hour. The samples were left on a shaker overnight in 95% ethanol at room temperature. We stained the embryos overnight at room temperature using a staining solution of Alcian blue 8GX (Sigma) and Alizarin red (Sigma) (staining solution: 5 ml of 0.4% Alcian blue in 70% ethanol, 5 ml glacial acetic acid, 70 ml of 95% ethanol, 20 ml of water; working solution: 100 μl of 0.5% Alizarin red (Sigma) added to 10 ml of staining solution). After staining overnight, we rinsed the embryos in water and dissolved soft tissues in 2% KOH in water for approximately eight hours. We then placed the samples in 0.25% KOH in water for 30 min and cleared tissues in increasing concentration of glycerol in 0.25% KOH in water. Samples were stored in 50% glycerol/0.25% KOH in water. Specimens were photographed with a scale using a Nikon SMZ1500 stereomicroscope linked to a Nikon D200 DSLR camera. We then took linear measurements of the bones using ImageJ v1.48 (*Schneider, Rasband & Eliceiri, 2012*).

## Histomorphometry

To analyze growth plate morphology and characterize any cellular disruption of endochondral ossification, we collected sixteen wildtype tibiae and eleven *Nabo* tibiae from mice at postnatal age P14. Tibiae were fixed in 4% paraformaldehyde (Sigma) for

4 to 5 days at 4 °C, and decalcified using Poly-NoCal (Polysciences, Inc). Proximal tibial growth plates were dehydrated, embedded in paraffin, and sectioned in the coronal plane at 5 μm. Sections were deparaffinized in xylene, stained with Mayer's hematoxylin (Sigma), 1% Alcian blue in 1% acetic acid (Sigma), 1% Phosphomolybdic acid (Sigma), 0.5% Sirius red (Direct Red 80, Sigma) in 0.5% acetic acid and then cover slipped. The stained sections were imaged at 5x using a digital microscope (Axioplan 2, Zeiss) with attached camera (Optronics) using StereoInvestigator v7. We selected a coronal section of the growth plate approximately halfway along the cranio-caudal axis for each specimen. In each section, we measured the total growth plate height (from epiphyseal to metaphyseal borders), resting/proliferative and hypertrophic zone heights at 5x magnification. For each variable, we repeated measurements 10 times evenly spaced across the section and took the average of these measurements. We identified the boundary between the zones based on the organization and size of the chondrocytes, as follows: the proliferative zone was considered as the zone with chondrocytes organized into columns and the resting zone as the zone proximal to the proliferative zone with sparse chondrocytes not organized in columns. The hypertrophic zone was determined based on the increased volume of the chondrocytes and reduced presence and staining of extracellular matrix. We also measured the height, in the direction of longitudinal growth, of at least ten terminal hypertrophic chondrocytes, adjacent to the chondro-osseous junction, at 20x magnification in nine *Nabo* mice and seven wildtype mice. Linear measurements were taken using the "Straight Line" tools in ImageJ v1.48 (*Schneider, Rasband & Eliceiri, 2012*). We calculated the means of the measurements taken in different regions of the growth plate for each individual before statistical analysis.

## Cell proliferation assay

To assess whether the dysplasia was associated with altered activity of the proliferative chondrocytes in the proximal growth plate, we injected three specimens of *Nabo* and eight wildtype mice intraperitoneally with 5-Bromo-2′-deoxyuridine (Sigma) in PBS at 50 μg/g of body weight (*Wojtowicz & Kee, 2006*). 5-Bromo-2′-deoxyuridine (BrdU) is incorporated into DNA during the S phase of the cell cycle (*Wojtowicz & Kee, 2006*). P14 mice were euthanized 24 h after BrdU injection. Tibiae were harvested, and fixed in 10% Neutral Buffered Formalin (Sigma) and decalcified using Cal-Ex$^{TM}$ II (Fisher Scientific). The proximal tibial growth plates of all specimens were dehydrated, embedded in paraffin, and sectioned in the coronal plane at 8 μm. Sections were deparaffinized in xylene, treated with 0.5% pepsin/0.05 MHCl (Sigma) and hydrogen peroxide (3% in methanol) to block endogenous peroxidase. Sections were then blocked with 10% goat serum/1%BSA/PBS for an hour at room temperature and incubated with rat monoclonal anti-BrdU antibody (1:100, AbD Serotec) overnight at 4 °C. After washes, the sections were incubated with biotinylated anti-rat IgG (1:200, Vector Laboratories) for an hour, and treated with streptavidin horseradish. Liquid DAB substrate kit was used to detect the BrdU signal (Invitrogen), and hematoxylin was used as a counterstain. We selected the coronal section of the growth plate closest to the center of the bone along the cranial-caudal axis for each specimen. Sections were imaged at 5x as above.
## Immunohistochemistry

To determine whether transcriptional regulation within the growth plate had been disrupted, we performed immunohistochemistry for Sox9 and Runx2, two transcription factors important for cell proliferation and hypertrophic differentiation, respectively. We used the same specimens used for the proliferation assay. Sections were deparaffinized in xylene, treated with hydrogen peroxide (15 min, 3% in methanol) to block endogenous peroxidase, and followed by antigen retrieval (30 min, 0.1% Triton-X). Sections were then blocked with normal serum for an hour at room temperature and incubated with anti-Sox9 primary antibody (1:300, AF3075, Bio-Techne) and anti-Runx2 primary antibody (1:400, ab192256, AbCam) overnight at 4 °C. After washes, the sections were incubated with HRP-conjugated secondary antibodies (Sox9: Bio-Techne HAF109, 1:100, Runx2: AbCam ab6721, 1:100) for an hour. Liquid DAB substrate kit (Invitrogen) was used to detect Sox9 and Runx2 and methyl green was used as a counterstain.

## Cortical and trabecular analysis

To determine whether *Nabo* have altered cortical and trabecular morphology in addition to their abnormal external skeletal morphology, we dissected and µCT scanned the right tibia from five P75 *Nabo* (three females and two males) and four age-matched wildtype (two females and males) (voxel size range 7.10–8.52, X-ray 75–78 KV; 100 µA, no filter). We first analyzed tibia diaphyseal cross-sectional properties, to determine whether cortical bone morphology differed between groups (*Nabo* and wildtype). Cross-sectional image stacks were imported into Fiji (ImageJ v1.50e) (*Schindelin et al., 2012*); we selected the tenth image, or approximately 80 µm, proximal to the tibia-fibula junction. We then used the plugin *BoneJ* (*Doube et al., 2010*) to measure the cross-sectional area (CSA), polar section modulus ($Z_{pol}$), and mean cortical thickness (MT) of the tibia. We calculated the index of robusticity (IR) of the bone by taking the polar section modulus ($Z_{pol}$) to the product of bone length and body mass (in mg$^{2/3}$) (*Cosman, Sparrow & Rolian, 2016*). Measurements of trabecular bone were performed in *BoneJ* in each tibia from the first image showing trabeculae distal to the partially fused growth plate, and extending 1 mm distally. We measured bone volume (BV), total volume (TV), bone volume fraction (BV/TV), trabecular thickness (Tb.Th), and tibia length as above.

## Statistical analysis

Measurements for the postnatal growth curve were analyzed using analysis of covariance (ANCOVA) with skull, clavicle, scapula, humerus, ulna, 5th metacarpal, femur, tibia, and 5th metatarsal lengths as dependent variables, line (*Nabo* vs wildtype) as categorical factors, and body mass as a covariate. Measurements of the skull elements at P75 were analyzed using ANCOVA using length and width of the elements as dependent variables and body mass as a covariate.

Cortical and trabecular bone variables were analyzed using ANOVA, with mouse line as categorical factor. Measurements of humerus, ulna and tibia length at E18 were analyzed using ANOVA. Measurements of the bone elements at E14 were analyzed using ANCOVA using humerus, ulna, radius, femur and tibia length as dependent variables and mean width of the eyes and width of nose as covariates.

Means from the histomorphometry data measurements were analyzed using ANCOVA with proximal growth plate height, the heights of the resting, proliferative and hypertrophic zones, and last hypertrophic chondrocyte height as dependent variables, line as categorial factor, and body mass as covariate.

All statistical analyses were performed using SPSS v23 (IBM Corp. Released 2015. IBM SPSS Statistics for Macintosh, Version 23.0. Armonk, NY: IBM Corp), and differences in means between the groups were considered statistically significant at $p < 0.05$.

## RESULTS

### The Nabo mice

*Nabo* mice are characterized by short, bowed forelimb and hind limb bones, with more pronounced shortening of the more proximal elements (stylopod and zeugopod) (Fig. 1). There is sometimes an extra thoracic vertebra accompanied by paired, or occasionally unilateral, ribs that express either as false ribs, or as true ribs with an extra sternal segment (Figs. 1C and 2B). *Nabo* mice are fertile, with viable litters. The litters are generally small, between 5 and 8 pups (vs. 12–15 for wildtype), and are weaned several days later than the wildtype mice due to the small size of the animals.

### Nabo postcranial phenotypes and growth

We took linear measurements of several elements of the skeleton to investigate how postnatal development contributes to the *Nabo* phenotype. We chose skeletal elements that develop by endochondral ossification (e.g., femur, tibia, ulna), intramembranous ossification (e.g., clavicle) or a combination of both (e.g., skull) and compared them to CD1 wildtype mice. Clavicle size does not differ during post-natal growth (Fig. 3, Table S1). Skull growth rates are also very similar, producing a skull that is approximately 5.3% shorter in *Nabo* than in the wildtype group at P125. Humerus, ulna, femur and tibia are significantly different across all post-natal ages, showing respectively 22.7%, 22.9%, 20.8% and 26.9% reduction in element size compared to age-matched wildtype mice at P125 (Fig. 3, Table S1). The scapula at P125 also shows a 25.5% reduction in element size compared to age-matched wildtype mice, and the length is significantly different for all post-natal stages except for P28 where the *p*-value is $p = 0.085$ (ANCOVA, see Table S1 for details) (Fig. 3). The autopod (hand and foot) is not affected as much as the stylopod and zeugopod: the fourth metacarpal and the metatarsal are shorter by 9.6% and 8.7% at P125. Metacarpal length is significantly different from wildtype only at P28 (ANCOVA, $F = 9.190$, $p = 0.019$) and P125 (ANCOVA, $F = 8.993$, $p = 0.024$) while metatarsal length is significantly different from wildtype at P75 (ANCOVA, $F = 12.147$, $p = 0,025$) and P100 (ANCOVA, $F = 9.414$, $p = 0.022$).

### Nabo craniofacial phenotype

μCT scans of *Nabo* skulls show no evident deformities such as tooth or jaw dysmorphologies, cleft palate, or other abnormal facial phenotypes (Fig. S2). Since the growth rate of the skull at age P75 is close to 0 (Fig. 3), we took linear measurements of several dorsal and ventral bones of the skull of five *Nabo* and five wildtype mice at

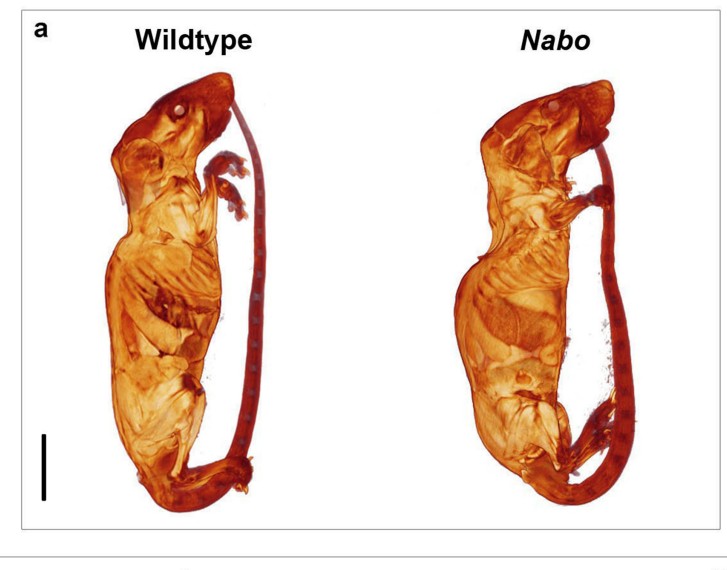

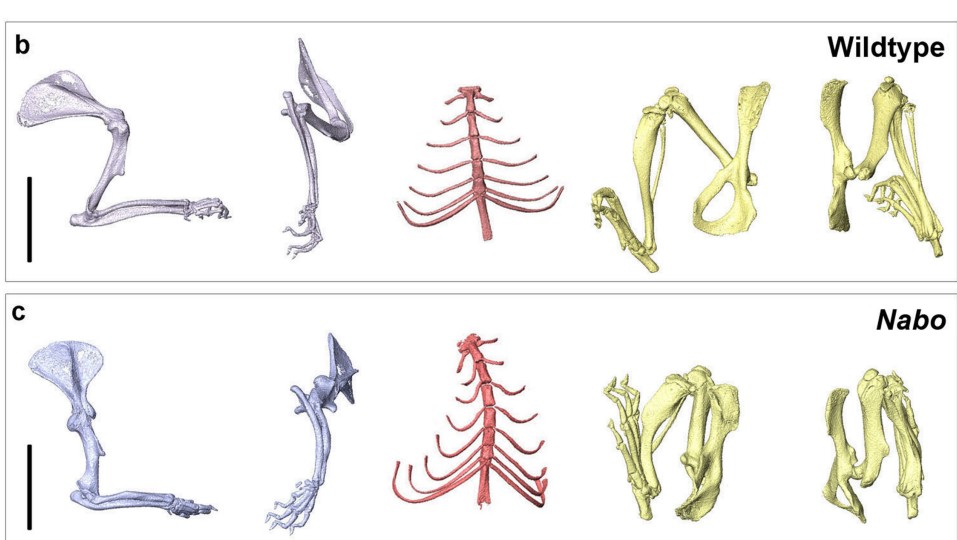

**Figure 1** *Nabo.* **adult morphology.** (A) Wildtype and *Nabo* mice skinned and stained with Lugol's iodine. The mice were μCT scanned and visualized with Amira v 5.4.2. (B, C) 3D surface models of forelimb, sternum and hind limbs in wildtype (B) and *Nabo* (C) derived from μCT scans. Scale bar = 10 mm.

postnatal age P75 (three males and two females each) from the μCT scans. We did not find any differences in the length of cranial vault bones that develop by intramembranous ossification, including parietals, frontals and nasal bones (*De Beer, 1937*) (Table 1, Fig. S2). The foramen magnum is significantly smaller in dorsoventral diameter by 6.5% (ANCOVA, $F = 6.046$, $p = 0.044$) and in width by 14% (ANCOVA, $F = 54.363$, $p < 0.001$). We also measured the length from the most dorsal landmark on the border of the foramen magnum to the most rostral landmark on the median internasal suture and found that the skull is significantly shorter than wildtype mice by 6.2% (ANCOVA, $F = 35.339$, $P = 0.001$), similar to the data from P125 mice (Table 1, Fig. S2). The measurements of the cranial base show that the occipital bone and sphenoid, which develop in part by

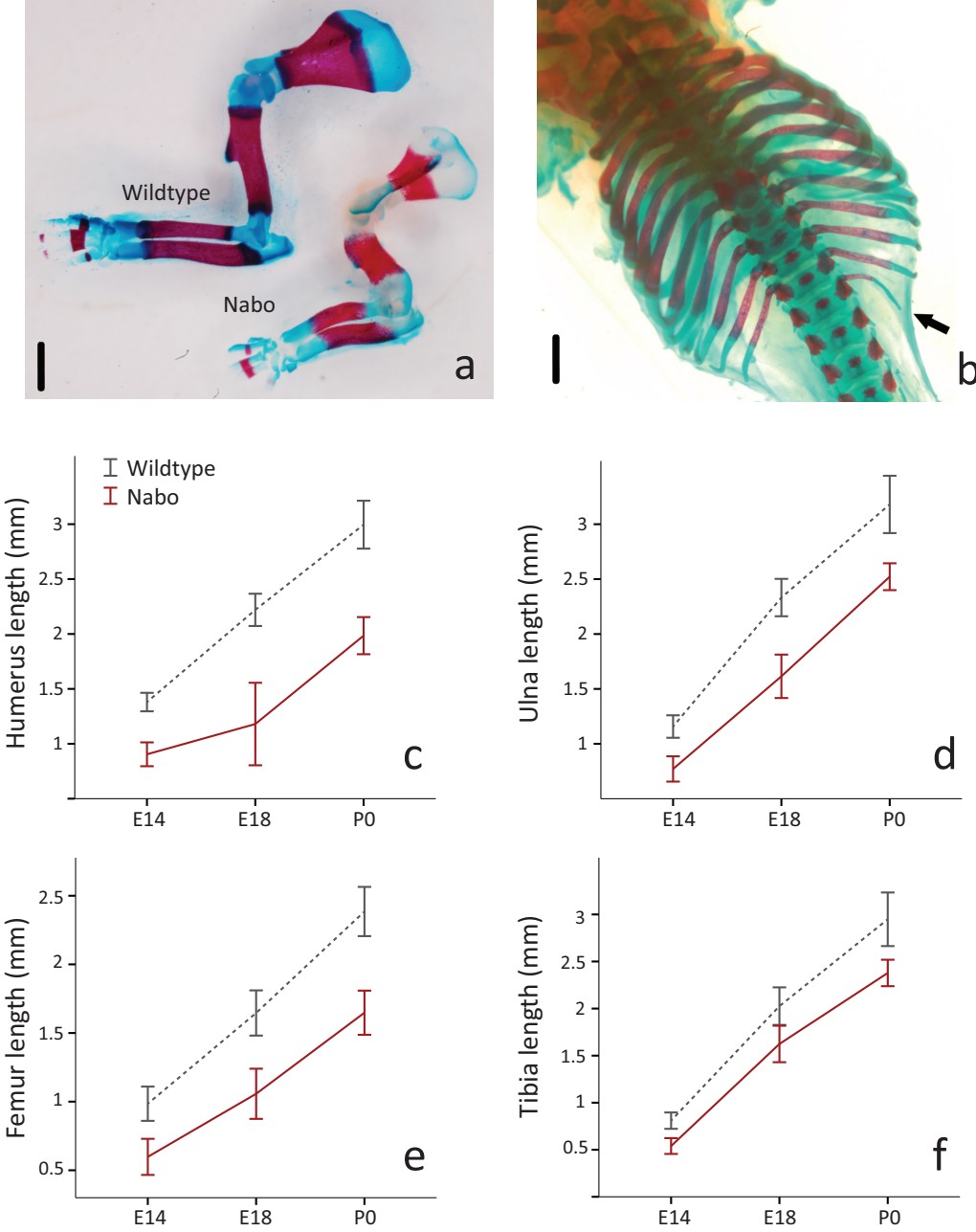

**Figure 2** **E18 fetal mice and prenatal growth trajectories.** (A, B) Samples stained using Alcian blue and Alizarin red. Scale bar = 1 mm. Wildtype and *Nabo* forelimbs of mice at developmental stage E18 (A). Example of extra rib expressed unilaterally in *Nabo* mice (black arrow, B). (C–F) prenatal growth of humerus (C), ulna (D), femur (E) and tibia (F) at developmental stage E14 and E18 and neonatal (P0). Wildtype: dashed grey lines and *Nabo:* solid red lines. Vertical bars indicate 95% confidence intervals.

endochondral ossification (*De Beer, 1937*), are reduced in size, with the former narrower by 21.2% (ANCOVA, $F = 69.294$, $p < 0.001$) and shorter by 19.7% (ANCOVA, $F = 125.170$, $p < 0.001$), and the latter shorter by 24.6% (ANCOVA, $F = 7.778$, $p = 0.0027$) (Table 1,

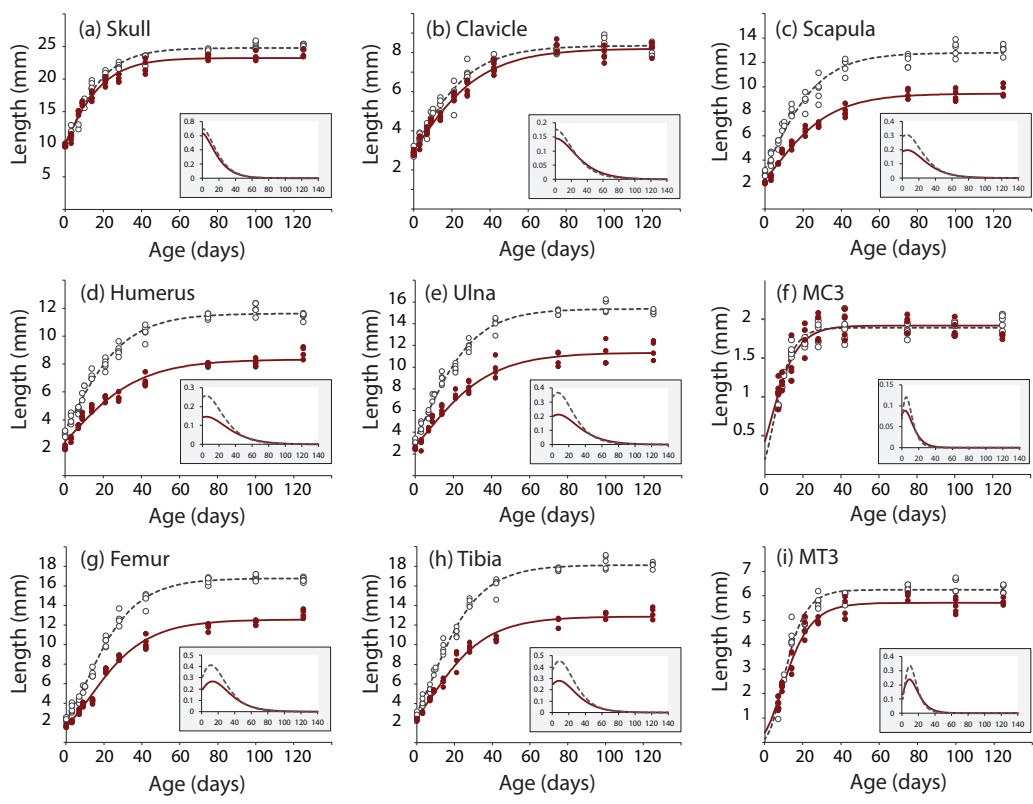

**Figure 3** **Postnatal growth curve for cranial (A), forelimb (B–F) and hind limb (G–I) skeletal elements in wildtype and *Nabo*.** The curves were obtained by fitting a Gompertz logistic growth function. Insets indicate growth rates, based on the first derivative of the Gompertz function. Wildtype: open circles, dashed lines, *Nabo:* red circle and lines.

Fig. S2). In comparison, the palatine, which develops by intramembranous ossification (*De Beer, 1937*), does not differ in size between *Nabo* and wildtype mice.

## Developmental timing of Nabo phenotype

To investigate the developmental timing of the appearance of the *Nabo* phenotype, we measured the size of the diaphysis of the tibia, humerus and ulna for *Nabo* and wildtype embryos at developmental stage E18. *Nabo* mice at this stage have a significantly shorter diaphysis in the humerus (46.8% shorter than wildtype) (ANOVA, $F = 119.463, p < 0.001$), ulna (30.7%) (ANOVA, $F = 58.191, p < 0.001$), femur (35.8%) (ANOVA, $F = 44.173, p < 0.001$), tibia (19.8%) (ANOVA, $F = 16.096, p = 0.004$), and in the body of the scapula (52.2%) (ANOVA, $F = 114.993, p < 0.001$) (Fig. 2 and Table 2). We then measured the length of the cartilage anlagen for the femur, tibia, humerus, ulna and radius of embryos stained with Lugol's solution at embryonic stage E14. Femur, tibia, humerus and ulna were shorter in *Nabo* mice compared to wildtype mice by respectively 39.3% (ANCOVA, $F = 6.521, p = 0.043$), 33.3% (ANCOVA, $F = 6.276, p = 0.046$), 34.5% (ANCOVA, $F = 24.695, p = 0.003$) and 33.4% (ANCOVA, $F = 10.935, p = 0.016$). Length differences remain present after controlling for covariation with internasal distance and interorbital

**Table 1** **Least squared means (mm) of linear measurements of skull elements using μCT scanning.**
Standard error presented in parenthesis. Asterisk and red font indicate statistical significant ($p < 0.05$).
Foramen magnum (FM). Df indicates degree of freedom. F indicates F-test (ANCOVA). p indicates p-value.

|  | Landmarks | Wildtype | *Nabo* | Statistics ANCOVA |
|---|---|---|---|---|
| Sample size |  | 5 | 5 |  |
| Body mass (g) |  | 37.92 | 31.23 |  |
| *Dorsal landmarks* |  |  |  |  |
| Parietal length | Land 1-2 | 4.429 (0.145) | 4.423 (0.145) | *Df* = 1; *F* = 0.001; *p* = 0.997 |
| Frontal length | Land 2-3 | 7.443 (0.144) | 7.349 (0.144) | *Df* = 1; *F* = 0.180; *p* = 0.684 |
| Nasal length | Land 3-4 | 8.659 (0.131) | 8.246 (0.131) | *Df* = 1; *F* = 4.293; *p* = 0.077 |
| FM height | Land 5-6 | **4.184 (0.072)** | **3.913 (0.072)*** | *Df* = 1; *F* = 6.046; *p* = 0.044 |
| FM width | Land 7-8 | **4.503 (0.056)** | **3.871 (0.056)*** | *Df* = 1; *F* = 54.363; *p* < 0.001 |
| Skull length | Land 4-5 | **24.738 (0.169)** | **23.207 (0.169)*** | *Df* = 1; *F* = 35.339; *p* = 0.001 |
| *Ventral landmarks* |  |  |  |  |
| Occipital length | Land 1-2 | **5.042 (0.098)** | **3.801 (0.098)*** | *Df* = 1; *F* = 125.170; *p* < 0.001 |
| Occipital width | Land 2-3 | **6.581 (0.082)** | **5.187 (0.082)*** | *Df* = 1; *F* = 69.294; *p* < 0.001 |
| Sphenoid length | Land 8-9 | **4.081 (0.189)** | **3.276 (0.189)*** | *Df* = 1; *F* = 7.778; *p* = 0.027 |
| Palatine length | Land 4-5 | 1.401 (0.143) | 1.204 (0.143) | *Df* = 1; *F* = 0.813; *p* = 0.397 |
| Palatine width | Land 6-7 | 3.633 (0.063) | 3.740 (0.063) | *Df* = 1; *F* = 1.239; *p* = 0.302 |

**Table 2** **Means (mm) of diaphyseal length of E18 *Nabo* mice and wildtypes.** Standard error presented
in parenthesis. Asterisk denotes significant differences in means ($p < 0.05$). *Df* indicates degree of freedom. F indicates F-test (ANOVA). p indicates p-value.

|  | Wildtype | *Nabo* | Statistics ANOVA |
|---|---|---|---|
| Sample size | 5 | 3 |  |
| Humerus | **2.22 (0.05)** | **1.18 (0.09)*** | *Df* = 1; *F* = 119.462; *p* < 0.001 |
| Scapula | **1.75 (0.05)** | **0.84 (0.07)*** | *Df* = 1; *F* = 114.993; *p* < 0.001 |
| Sample size | 5 | 5 |  |
| Femur | **1.65** | **1.06** | *Df* = 1; *F* = 44.173; *p* < 0.001 |
| Tibia | **2.03 (0.07)** | **1.62 (0.07)*** | *Df* = 1; *F* = 16.096; *p* = 0.004 |
| Ulna | **2.33 (0.06)** | **1.61 (0.07)*** | *Df* = 1; *F* = 58.191; *p* < 0.001 |

distance (see Table 3 for details, Fig. S3). The histology of the limb at embryonic stage E14.5
shows that the tibial anlagen is not only shorter in *Nabo*, but also lacks a differentiated
hypertrophic chondrocyte zone when compared with wildtype (Fig. S4).

## Nabo growth plate structure and chondrocyte dynamics

We analyzed the proximal tibia growth plates of 14-day old *Nabo* mice and compared them
to age-matched wildtype mice because at this stage growth rate in the tibia is close to its
maximum, and therefore, we would expect differences in growth plate structure to be most
apparent (Fig. 3). Overall, the total growth plate was significantly taller than wildtype, by
31.6% (ANCOVA, $F = 33.434$, $p < 0.001$). Similarly, the proliferative and hypertrophic
zones were significantly taller than wildtype, respectively by 42.8% (ANCOVA, $F = 24.363$,
$p < 0.001$) and by 25.0% (ANCOVA, $F = 22.586$, $p < 0.001$). The resting zone showed no

**Table 3** **Least squared means (μm) of cartilage anlagen size of E14 *Nabo* mice and wiltypes.** Mean of eyes and nose width were used as covariates for statistical analysis. Standard error presented in parenthesis. Asterisk and red font denote significant differences in means ($p < 0.05$). Df indicates degree of freedom. *F* indicates *F*-test (ANCOVA). *p* indicates *p*-value.

| | Wildtype | *Nabo* | Statistics ANCOVA |
|---|---|---|---|
| Sample size | 6 | 5 | |
| Mean eyes width (μm) | 802 | 704 | |
| Mean nose width (μm) | 573 | 550 | |
| Femur (μm) | **894 (56)** | **646 (56)*** | *Df* $= 1$; *F* $= 6.521$; *p* $= 0.043$ |
| Tibia (μm) | **745 (38)** | **580 (38)** * | *Df* $= 1$; *F* $= 6.276$; *p* $= 0.046$ |
| Humerus (μm) | **1330 (46)** | **931 (46)*** | *Df* $= 1$; *F* $= 24.695$; *p* $= 0.003$ |
| Ulna (μm) | **1096 (50)** | **805 (50)*** | *Df* $= 1$; *F* $= 10.935$; *p* $= 0.016$ |

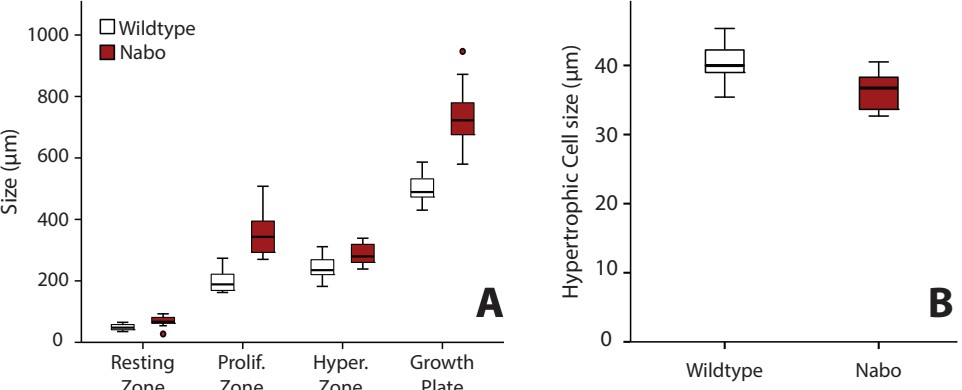

**Figure 4** **Growth plate histomorphometry.** (A) Box plots of resting, proliferative, hypertrophic zones, and total proximal tibial growth plate size. (B) Box plots of terminal hypertrophic chondrocyte height. Wildtype in grey; *Nabo* in red. Box represents the 25th and 75th percentile; black bars within boxes represent the median; whisker represents the non-outlier range and dots represent outliers.

difference in height (Fig. 4A, Table 4). The proximodistal height of the last hypertrophic chondrocyte adjacent to the chondro-osseous junction (COJ) is shorter in *Nabo* mice compared to wildtype mice by 12.1% (ANCOVA, $F = 6.454$, $p = 0.025$) (Fig. 4B, Table 4).

The growth plate of *Nabo* shows defects at the COJ but not in the general columnar organization of chondrocytes (Fig. 5). The trabecular bone in the metaphysis lacks clear longitudinal organization of the trabeculae and is generally more sparse than in wildtype mice (see below). The COJ is separated from the hypertrophic zone by a zone rich in bone marrow cells. Many specimens show cartilage matrix, as indicated by the presence of Alcian stain, with apparently healthy chondrocytes partially or completely enclosed within trabecular bone (Fig. 5C).

We performed a cell proliferation assay to determine whether proliferative chondrocytes were actively dividing in the growth plates of *Nabo* mice. BrdU positive cells were present in the growth plate, suggesting active chondrocyte proliferation. However, in contrast to wildtype mice, BrdU positive cells were mostly concentrated in the resting zone and in

**Table 4  Least squared means of growth plate zone size using histomorphometry.** Standard error presented in parentheses. Asterisk denotes significant differences in means ($p < 0.05$). $Df$ indicates degree of freedom. $F$ indicates $F$-test (ANCOVA). $p$ indicates $p$-value.

| | Wildtype | *Nabo* | Statistics ANCOVA |
|---|---|---|---|
| *Growth Plate Zones* | | | |
| Sample size | 16 | 11 | |
| Body mass (g) | 8.35 | 6.54 | |
| Total growth plate (μm) | **503.00 (22.28)** | **734.97 (28.19)**★ | $Df = 1; F = 33.434; p < 0.001$ |
| Resting zone (μm) | 53.77 (3.61) | 61.72 (4.56) | $Df = 1; F = 1.501; p = 0.232$ |
| Proliferative zone (μm) | **201.22 (16.93)** | **351.658 (21.41)**★ | $Df = 1; F = 24.362; p < 0.001$ |
| Hypertrophic zone (μm) | **229.10 (8.94)** | **305.62 (11.31)**★ | $Df = 1; F = 22.586; p < 0.001$ |
| *Last Hypertrophic Chondrocyte Height* | | | |
| Sample size | 7 | 9 | |
| Body mass (g) | 8.22 | 6.53 | |
| Last hypertrophic chondrocyte height (μm) | **40.83 (1.34)** | **35.88 (1.15)**★ | $Df = 1; F = 6.454; p = 0.025$ |

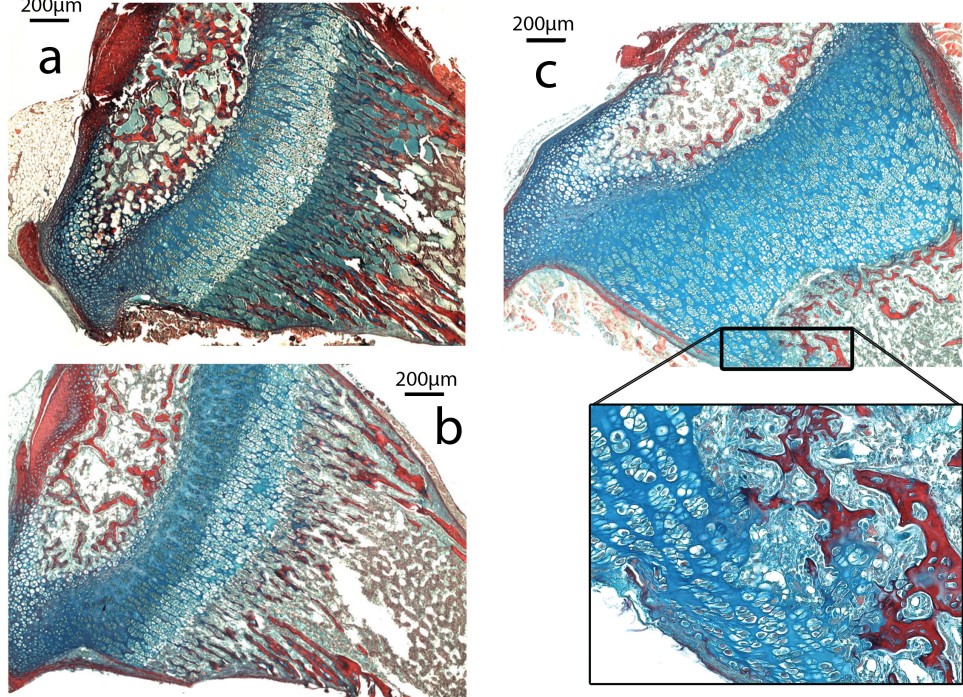

**Figure 5  Histomorphometry.** (A–C). Histology of proximal growth plate tibia of 14 days old wildtype (A), heterozygote (B) and *Nabo* (C). Inset illustrates the chondro-osseous junction in *Nabo* growth plate.

the most proximal proliferative chondrocytes (Figs. 6A and 6B). Immunohistochemistry of Sox9 mirrors the BrdU results, with Sox9 positive cells restricted to the most proximal proliferative zone (Figs. 6C and 6D). Immunohistochemistry of Runx2 shows positive cells in the presumptive pre-hypertrophic zone (Figs. 6E and 6G) and in the metaphysis (Fig. 6F and 6H) suggesting that the phenotype is not caused by an absence of osteoblasts.

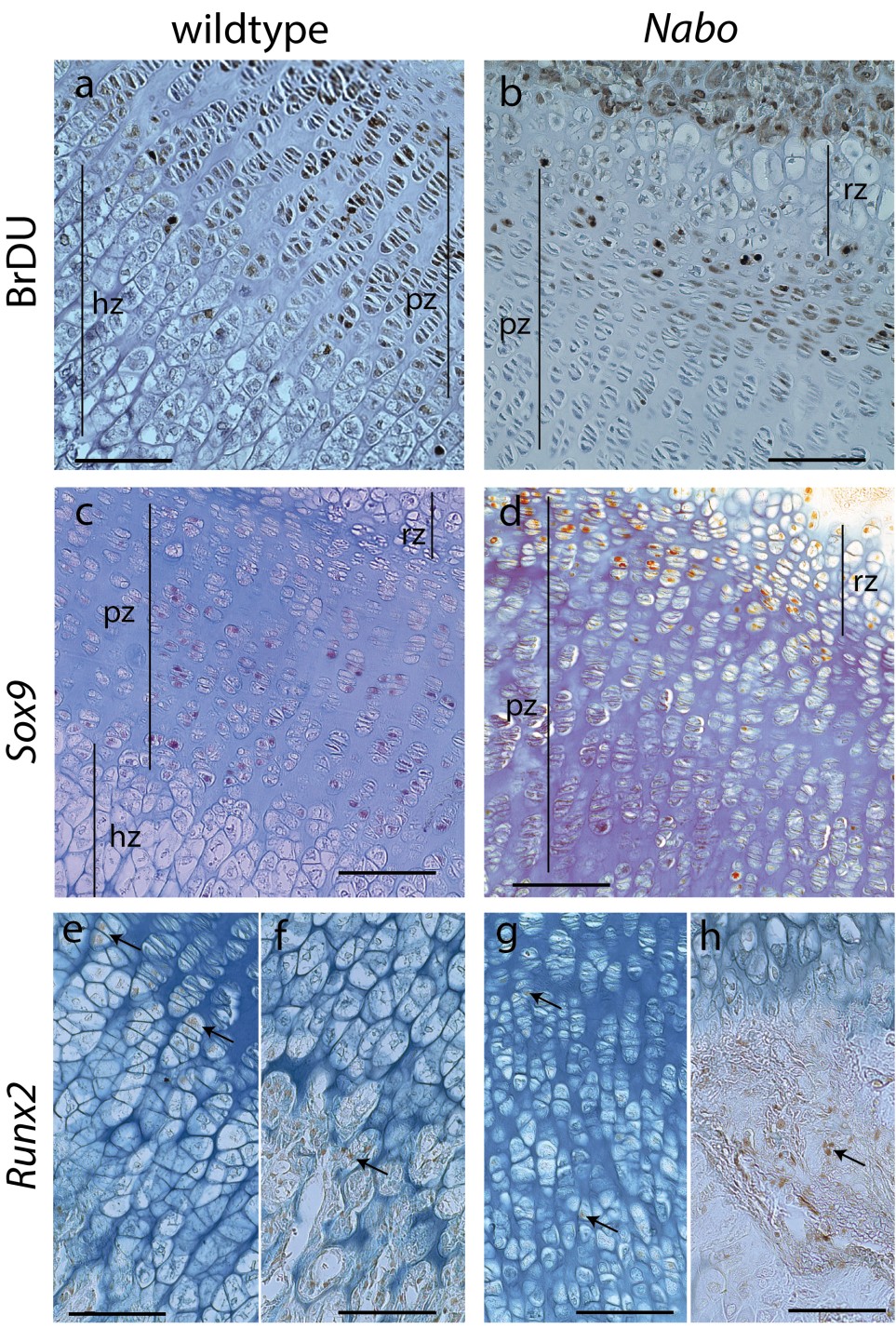

**Figure 6** **Cell Proliferation and Immunostaining Assays.** Growth plate cell proliferation and immunostaining assays of proximal growth plate tibiae of 14 days old mice. (A, B) BrdU staining of wildtype (A) and *Nabo* (B), showing decreased mitotic activity in the proliferating zone (pz) in *Nabo* compared with wildtype. (C, D) immunostaining against Sox9 in growth plates of wildtype (C) and *Nabo* (D). Sox9 protein shows comparable distribution to BrdU staining, with increased staining in the upper proliferative zone and resting zone (rz) relative to the lower pz in *Nabo* (E–H) immunostaining against Runx2 in the growth plate and metaphysis of wildtype (E, F) and *Nabo* (G, H). The antibody is detected in both samples at the transition zone between proliferating and hypertrophic chondrocytes (arrows in e,g) and in osteoblasts in trabecular bone (arrows in F, H). Scale bar − 100 μm.

**Table 5  Means of cortical and trabecular bone variables.** Standard error presented in parentheses. Asterisk denotes significant differences in means ($p < 0.05$). *Df* indicates degree of freedom. *F* indicates *F*-test (ANCOVA). *p* indicates *p*-value.

| | Wildtype | *Nabo* | Statistics ANCOVA |
|---|---|---|---|
| Sample size | 4 | 5 | |
| Body mass (mg) | 37.04 | 27.81 | |
| Tibia length (mm) | 17.03 | 11.28 | |
| Bone Volume (BV, mm$^3$) | **0.97 (0.15)** | **0.56 (0.05)\*** | $Df = 1$; $F = 7.737$; $p = 0.027$ |
| Total Volume (TV, mm$^3$) | 2.71 (0.29) | 2.20 (0.21) | $Df = 1$; $F = 2.082$; $p = 0.192$ |
| Bone Volume Fraction (BV/TV) | **0.35 (0.19)** | **0.25 (0.06)\*** | $Df = 1$; $F = 29.048$; $p = 0.001$ |
| Trabeculae Thickness (Tb.Th, mm) | **0.060 (0.001)** | **0.048 (0.001)\*** | $Df = 1$; $F = 46.517$; $p < 0.001$ |
| Cross-sectional Area (CSA, mm$^2$) | 0.95 (0.09) | 1.02 (0.06) | $Df = 1$; $F = 1.501$; $p = 0.232$ |
| Polar Section Modulus ($Z_{pol}$, mm$^3$) | 0.29 (0.03) | 0.37 (0.02) | $Df = 1$; $F = 3.867$; $p = 0.090$ |
| Robusticity (IR, mm$^2$ mg$^{-2/3}$) | **$1.5 \times 10^{-5}$ ($0.1 \times 10^{-5}$)** | **$3.6 \times 10^{-5}$ ($0.2 \times 10^{-5}$)\*** | $Df = 1$; $F = 77.807$; $p < 0.001$ |
| Mean Thickness (MT, mm) | 0.36 (0.03) | 0.35 (0.02) | $Df = 1$; $F = 0.275$; $p = 0.616$ |

## Nabo trabecular architecture

In histology slides of proximal tibiae of 14-day-old pups, we observed sparse trabecular bone in *Nabo* in comparison with age-matched wildtypes (see above, and Fig. 5). We then investigated whether the trabecular bone in the groups was similar at skeletal maturity. We μCT scanned tibiae for each group ($n = 4 - 5$ each) and analyzed the proximal trabecular bone and the cross-sectional properties using BoneJ (18,24). Bone volume (BV) was significantly lower in *Nabo* mice than in wildtype by 42.3% (ANCOVA, $F = 7.737$, $p = 0.027$), and, because total endosteal volume (TV) was only 18.8% smaller in Nabo, this produced a bone volume ratio (BV/TV) that was also significantly lower in *Nabo* mice, by 28.6% compared to wildtype (ANCOVA, $F = 29.048$, $p = 0.001$). The trabeculae in *Nabo* mice are also thinner by 20% compared to the wildtype group (Tb.Th, ANCOVA, $F = 46.517$, $p < 0.001$) (Table 5). These adult data are consistent with our early postnatal histological observations (Fig. 5).

We measured the cross-sectional area (CSA), mean thickness (MT) of the diaphyseal cortex, and the polar section modulus ($Z_{pol}$), a measure of the distribution of bone from the central axis of the bone diaphysis (*Young et al., 2014*). $Z_{pol}$ is 21.6% higher in *Nabo* than in wildtype, but we did not find statistical differences in any these measurements between the groups. We also calculated the index of robusticity (IR) as polar section modulus scaled to body mass and tibia length. IR can be used as a proxy for the bending strength of the tibia of each mouse (*Young et al., 2014*). IR is 58% higher in *Nabo* compared to the wildtype group (ANCOVA, $F = 77.807$, $p < 0.001$), which suggests that, at least from a bone geometry perspective, *Nabo* mice are more resistant to bending loads (Table 5).

## DISCUSSION

### Nabo mice show a previously uncharacterized SD-like phenotype

The *Nabo* mice present a peculiar skeletal phenotype that, based on the pedigree (Fig. S1), appears to be autosomal recessive. The phenotype affects proximal limb elements more strongly than autopods (Fig. 1). Additionally, *Nabo* mice exhibit a pronounced kyphosis

and frequent extra thoracic vertebrae (Figs. 1C and 2B). Because we did not find size differences in bones that develop via intramembranous ossification (such as the clavicle and palatine bone in the skull) (*De Beer, 1937*), we tentatively conclude that the phenotype derives primarily from differences in cell and molecular mechanisms of endochondral ossification.

The *Nabo* phenotype shares certain features with SD-like phenotypes that disrupt normal bone formation and development, such as short bowed limbs, kyphosis, and vertebral anomalies (*Hurst, Firth & Smithson, 2005*). The growth plate also shows irregularities in bone and cartilage matrix at the COJ that may contribute to this phenotype. However, there are several major differences between the *Nabo* phenotype and known forms of SD (Table S2). Notably, *Nabo* mice lack severe head and facial dysmorphologies and other clinical symptoms (including brachydactyly or growth plate disorganization) that characterize many SDs (*Krakow & Rimoin, 2010*; *Hurst, Firth & Smithson, 2005*). This suggests that *Nabo* is not caused by mutations in major signaling pathways that involve FGFs, BMPs and Wnt, or matrix proteins such as collagen and aggrecan (*Krakow & Rimoin, 2010*; *Rimoin et al., 2007*), which often leads to craniofacial phenotypes.

We surveyed the Online Mendelian Inheritance in Man (OMIM) database (http://www.omim.org) to identify potential human disorders with similarities to the *Nabo* phenotype. Using the keyword "bowed" as our search term, we found 238 results, of which three skeletal dysplasias with unknown etiologies presented similar clinical signs to our phenotype (Fig. 7): spondylometaphyseal dysplasia (OMIM 607543) (*Kozlowski & Poon, 2003*), metaphyseal acroscyphodysplasia (OMIM 250215) (*Verloes et al., 1991*) and metaphyseal dysostosis (OMIM 250420) (*Rimoin & McAlister, 1971*). The remaining disorders differ significantly from our phenotype because they exhibit severe craniofacial defects, fall under the category of *osteogenesis imperfecta*, had no short limb phenotype, only showed a subset of the *Nabo* phenotype defects, or show other defects not seen in the *Nabo* phenotype. Based on these results, we may tentatively exclude most of the known genetic causes of well-characterized SDs phenotypes (*Krakow & Rimoin, 2010*; *Ballock & O'Keefe, 2003*; *Hurst, Firth & Smithson, 2005*) as the basis of the *Nabo* mutation.

## The Nabo phenotype may involve dysregulation of limb initiation and patterning

Our data show that mice at embryonic stage E14 already demonstrate the *Nabo* phenotype, indicating that this phenotype must appear earlier in development. At E14, the stylopod and zeugopod have already formed as cartilage anlagen while the autopod is still mostly composed of undifferentiated mesenchymal cells (*Taher et al., 2011*). Because more proximal limb elements are more strongly affected in *Nabo*, and because the frequent extra thoracic vertebrae suggest dysregulation of somitogenesis, the *Nabo* phenotype may appear as early as limb initiation and patterning. This is further supported by our observations of incomplete differentiation of chondrocytes in both the tibial anlagen at embryonic stage E14.5 and within the postnatal growth plate in *Nabo*, suggesting that *Nabo* chondrocytes do not differentiate as in the wildtype mice.

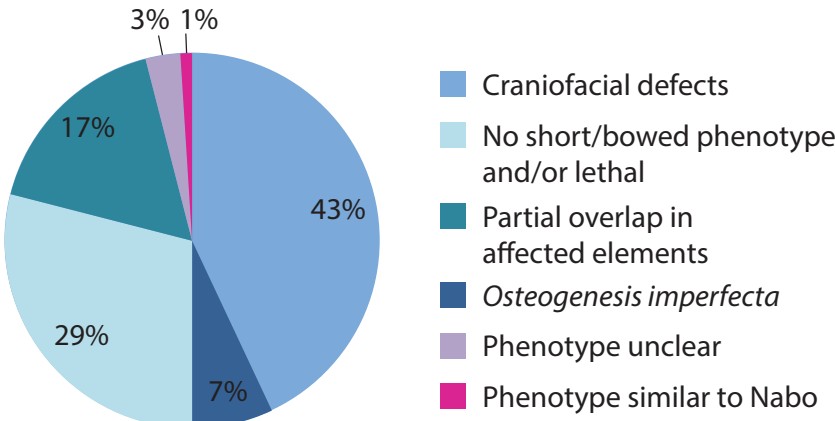

**Figure 7** **OMIM Skeletal Dysplasias.** Pie chart of the 238 diseases resulting from the search in the OMIM database using the keyword "bowed" (www.omim.org). Dark pink represents the percentage of diseases with *Nabo*-like phenotype. Light pink represents the percentage of diseases with unclear phenotype. Green represents the percentage of diseases where only some elements of the limbs were affected. Light green represents the percentage of diseases without short bowed limbs, or which are embryonic/perinatal lethal. Dark blue represents the percentage of diseases associated with *osteogenesis imperfecta*. Light blue represents the percentage of diseases that are characterized by craniofacial defects as well as limb bone defects.

The presence of the extra thoracic vertebra (if it is the same mutation) also suggests a possible gradient regulation or specific time point dysregulation during somitogenesis (*Dubrulle & Pourquié, 2004*). The number of lumbar, sacral and caudal vertebrae does not differ, with extra vertebrae appearing only in the thoracic region (Fig. 2).

We have identified differences in Sox9 protein distribution in the tibial growth plate, but the mechanism behind the *Nabo* phenotype is still unclear. Gene expression analysis and genotyping will be essential to narrow down the genetic causes of this phenotype, and to understand how the cause of this rare variant fits into the skeletal development program. Further investigation of this unique mutant line has the potential to shed light on poorly-understood but critical aspects of limb development not accounted for by the major signaling pathways which have been the focus of most research to date. *Nabo* may share a genetic basis with one or more rare skeletal dysplasias, making it a promising model to understand these disorders as well.

## CONCLUSIONS

The *Nabo* mutation represents a novel type of skeletal dysplasia with specific defects consistent with a dysregulation of endochondral ossification. This mutation is autosomal recessive with viable homozygotes facilitating ease of establishing and maintaining mutant lines. *Nabo* mice show short bowed limbs and mild axial defects as well as changes in growth plate morphology and reduction in trabecular bone, but show normal formation of intramembraneous skeletal elements. The novel phenotype and specificity of the defects observed suggests that the *Nabo* mouse may provide novel insights into less-understood aspects of endochondral ossification as well as skeletal dysplasias more broadly.

**List of abbreviations**

| | |
|---|---|
| **BrdU** | Bromo-2′-deoxyuridine |
| **BV** | bone volume |
| **BV/TV** | bone volume fraction (i.e., bone volume/total volume) |
| **CSA** | cross-sectional area |
| **COJ** | chondro-osseous junction |
| **IR** | index of robusticity |
| **MT** | mean thickness of the diaphyseal cortex |
| **μCT** | micro-computed tomography |
| **OMIM** | Online Mendelian Inheritance in Man |
| **SD** | Skeletal dysplasia |
| **Tb.Th** | trabecular thickness |
| **TV** | total volume |
| $\mathbf{Z_{pol}}$ | polar section modulus |

## ACKNOWLEDGEMENTS

The authors thank the staff at the University of Calgary Health Sciences Animal Resource Center for the care provided to our mouse samples. Thanks also to John Matyas and Dragana Ponjevic for assistance with histological analyses, Jason Anderson and Jessica Theodor for access to the SkyScan uCT scanner, and Alexandra Dowhanik for E18 wildtype samples. Rebecca Green, David Katz and Jason Pardo provided feedback on earlier versions of the manuscript, and their help is gratefully acknowledged.

### Funding

Marta Marchini and Campbell Rolian were supported by the University of Calgary Faculty of Veterinary Medicine. Campbell Rolian was also supported by Discovery Grant 4181932 from the Natural Sciences and Engineering Research Council of Canada. There was no additional external funding received for this study. The funders had no role in study design, data collection and analysis, decision to publish, or preparation of the manuscript.

### Grant Disclosures

The following grant information was disclosed by the authors:
University of Calgary Faculty of Veterinary Medicine.
Discovery: 4181932.

### Competing Interests

The authors declare there are no competing interests.

## Author Contributions

- Marta Marchini conceived and designed the experiments, performed the experiments, analyzed the data, prepared figures and/or tables, authored or reviewed drafts of the paper, approved the final draft.
- Elizabeth Silva Hernandez performed the experiments, authored or reviewed drafts of the paper, approved the final draft.
- Campbell Rolian conceived and designed the experiments, analyzed the data, contributed reagents/materials/analysis tools, prepared figures and/or tables, authored or reviewed drafts of the paper, approved the final draft.

## Animal Ethics

The following information was supplied relating to ethical approvals (i.e., approving body and any reference numbers):

The experiments were approved by the Health Sciences Animal Care Committee at the University of Calgary under the protocol number: AC13-0077.

## Data Availability

Morphometric measurements used in all statistical analyses are available in Data S1. Measurements were collected from micro-CT image stacks (JPEG format), which are available from the authors upon request.

## Supplemental Information

Supplemental information for this article can be found online at http://dx.doi.org/10.7717/peerj.7180#supplemental-information.

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
