# Peer review of "Morphology and development of a novel murine skeletal dysplasia"

_PeerJ, doi:10.7717/peerj.7180_

## Round 0.1 · original submission · Major Revisions

I have received reviews from two researchers with expertise in analyzing growth plate and skeletal abnormalities in mice. Both commend the paper on the relevance and quality of the analysis and the clarity of the manuscript. One reviewer suggests minor changes while the second suggests more substantial additions. While I do not think a full mechanisms underlying the SD is necessary at this time, but I have indicated major revisions because I would like to hear your response to the well informed reviewer's suggestions and concerns.

I also have some specific edits:

Line 79: Replace mouse with mice or say "in a mouse model" perhaps.

Line 93: Please provide a brief comment on whether this is an inbred or outbred strain.

Line 122: Add a comma between P9/10 P14.

Line 256: Extra period.

Line 346-348: Commas around "by 42.3%" and "by 28.6%" seem extraneous.

Line 350: Extra period.

I look forward to your revised manuscript and response to the reviewers.

Reviewer 1 ·

Basic reporting

In this manuscript entitled “Morphology and development of a novel murine skeletal dysplasia”. The goals of the study are clearly defined at the outset of the study and the authors organized the work to clearly achieve these goals. Data in figures is organized, easy to follow and relevant to the presented hypothesis.

Experimental design

The authors describe the phenotype of a novel mutant that arose naturally in the lab from a related ongoing study of the Longshanks mouse line. The materials and methods are thorough in describing the evolution of this project; important to the background for the study. The extensive skeletal analyses via microCT and other measurements shows a clear shortened limb phenotype of these mice that may allow for novel insight once the mutation has been identified in future studies. I commend the authors on this careful analysis through the postnatal growth of the animals.

Validity of the findings

Based on the data presented, the authors describe a mutation that negatively impacts endochondral ossification, rather than intramembranous ossification. These data are presented in a compelling manner by comparing the size and structure of bones generated by these two processes. The study, however, falls short of more detailed studies that would provide more substantial evidence that this is the case and that will be critical in future studies of these animals. The following points should be addressed prior to publication:

1. Based on histologic evidence in Figure 5, it is clear that chondrocytes are severely affected by the Nabo mutation during postnatal skeletal development. The authors also describe shortened limbs during embryonic development in Figure 3 and Table 2. Given that proper establishment of the cartilage anlagen at embryonic stages is critical to formation of the postnatal growth plate, the authors should also include histologic images of the developing skeleton at embryonic stages (similar to those provided in Figure 5).

2. The combined data of this report strongly suggests that chondrocytes are most affected by the Nabo mouse mutation, however the authors fall short in providing more mechanistic evidence that this is the case, especially at embryonic stages of development. Further histologic studies would allow the authors to conclude more clearly that chondrocytes (and endochondral ossification) are chiefly affected. Possible analyses could include in situ or immunohistochemistry for CollagenX, Collagen2, and Aggrecan.

3. In line 373 and 374, the authors suggest that the growth plates of Nabo mice “show irregularities in bone and cartilage matrix at the chondrosseous junction.” A few mechanistic studies should be provided to support this point further; the only evidence provided is histologic images of these regions. In situ or immunostaining for CollagenX, Collagen2, MMP13, or similar would provide clearer evidence that matrix is affected in the junction (i.e., lack of collagenX expression would suggest defects in chondrocyte maturation which lead to the disruption of the chondro-osseous junction). TRAP staining for osteoclasts may also provide the authors with evidence that matrix is being degraded properly for endochondral ossification.

·

Basic reporting

No comment

Experimental design

Please provide a rationale for the ages and sample sizes chosen for each analysis. For example, skulls of N= 5 mice per genotype were examined by microCT at postnatal day 75, but growth plate histology was done on only N=3 Nabo mice but N=8 wild type at postnatal day 14 (lines 132 and 174). Why not examine growth plates of older mice (5 weeks and/or 8 weeks age) and why not examine more than N=3 of the Nabo? Please provide justification. Was a power analysis conducted to determine sample sizes?

Validity of the findings

Minor comment: figures 2 and 3 are referred to out of order

Additional comments

This manuscript thoroughly characterizes a skeletal dysplasia-like phenotype in a new mouse mutation. The paper is organized and well-written. Although it does not appear to mimic any known (published) human skeletal condition, this new model may have important implications for understanding mechanisms of endochondral ossification during normal and aberrant growth. My only substantive concerns relate to the ages and sample sizes as described in the Experimental Design section of this review. There needs to be better justification for why each time point was selected and why sample sizes vary between analyses.

I also recommend changing all instances of growth plate "width" to "height" to match more customary terminology since it is measured in the direction of longitudinal growth.

---

## Round 0.2 · Minor Revisions

Thank you for submitting your revised manuscript. To my eye you have sufficiently addressed the reviewers' concerns. I have re-read the manuscript and still see a number of minor typos and errors in punctuation. I have uploaded a pdf with suggested edits. I also noticed that Spondylo-epiphyseal dysplasia and Hypophosphatasia are misspelled in Table S2. Please address these and resubmit before the manuscript is officially accepted and sent to on to make the proofs. Congratulations on your very nice work.

---

## Round 0.3 · accepted · Accept

Thank you for your last revisions on the manuscript. I hope you agree that the review process has improved this very interesting study.

#